# Sclerostin inhibits interleukin-1β-induced late stage chondrogenic differentiation through downregulation of Wnt/β-catenin signaling pathway

**Kazuma Miyatake, Ken Kumagai**[ID]*, **Sosuke Imai, Yasuteru Yamaguchi, Yutaka Inaba**

Department of Orthopaedic Surgery and Musculoskeletal Science, Graduate School of Medicine, Yokohama City University, Yokohama, Japan

* kumagai@yokohama-cu.ac.jp

## Abstract

It is known that Wnt/β-catenin signaling induces endochondral ossification and plays a significant role in the pathophysiology of osteoarthritis (OA). Sclerostin is a potent inhibitor of the Wnt/β-catenin signaling pathway. This study investigated the role of sclerostin in the endochondral differentiation under an OA-like condition induced by proinflammatory cytokines. ATDC5 cells were used to investigate chondrogenic differentiation and terminal calcification, and 10 ng/ml IL-1β and/or 200 ng/ml sclerostin were added to the culture medium. IL-1β impaired early chondrogenesis from undifferentiated state into proliferative chondrocytes, and it was not altered by sclerostin. IL-1β induced progression of chondrogenic differentiation in the late stage and promoted terminal calcification. These processes were inhibited by sclerostin and chondrogenic phenotype was restored. In addition, sclerostin restored IL-1β-induced upregulation of Wnt/β-catenin signaling in the late stage. This study provides insights into the possible role of sclerostin in the chondrogenic differentiation under the IL-1β-induced OA-like environment. Suppression of Wnt signaling by an antagonist may play a key role in the maintenance of articular homeostasis and has a potential to prevent the progression of OA. Thus, sclerostin is a candidate treatment option for OA.

## Introduction

Osteoarthritis (OA) is a degenerative joint disease, characterized by cartilage degradation, subchondral bone sclerosis, osteophyte formation, and synovial inflammation. A complex network of multifactorial mechanisms including biochemical, mechanical, and enzymatic aspects are involved in the pathogenesis of OA [1]. Proinflammatory cytokines such as interleukin (IL)-1β, IL-6, and tumor necrosis factor (TNF)-α are the critical mediators of the disturbed processes implicated in OA pathophysiology [2].

The Wnt/β-catenin signaling pathway plays a significant role in the pathophysiology of OA [3]. A previous study demonstrated that inhibition of Wnt/β-catenin signaling by small molecules can effectively prevent IL-1β- and TNFα-induced cartilage degradation by blocking the

**Data Availability Statement:** All relevant data are within the manuscript and its Supporting Information files.

**Funding:** This work was supported by a Grant-in-Aid for Scientific Research from the Ministry of Education, Culture, Sports, Science and Technology of Japan (#25462347).

**Competing interests:** The authors have declared that no competing interests exist.

production of matrix metalloproteinase (MMP) [4]. Furthermore, tissue-specific activation of Wnt/β-catenin signaling in articular chondrocytes of adult mice resulted in progressive loss of articular cartilage and an OA-like phenotype [5]. Thus, blockade of Wnt/β-catenin signaling may be proposed as a therapeutic target.

Sclerostin, encoded by the *SOST* gene, is known to be one of the Wnt signaling antagonists [6]. Sclerostin regulates disease processes in OA by opposing the effects of promotion of disease-associated subchondral bone sclerosis, while inhibiting the degradation of cartilage [7]. The deficiency of SOST aggravates the OA phenotype by increasing catabolic activity of cartilage [8], and SOST-knockout mice exhibited severe progression of OA in response to joint instability, suggesting that sclerostin may contribute to the maintenance of cartilage integrity in OA [9]. However, effects of sclerostin on terminal calcification of chondrocytes in the osteoarthritic environment are unknown and need to be elucidated, considering that endochondral ossification signals may be important for OA progression [10].

We previously demonstrated that *SOST* is upregulated in the early stage of chondrogenic differentiation, but is not required for endochondral ossification [11]. This study focused on the role of sclerostin in the chondrogenic differentiation under the OA-like condition induced by proinflammatory cytokines. We hypothesized that sclerostin upregulates chondrogenic differentiation to proliferating chondrocytes and downregulates endochondral ossification under the proinflammatory cytokine-induced condition. This study investigated IL-1β-induced osteochondral differentiation *in vitro*, and examined whether sclerostin can restore the chondrogenic phenotype.

## Materials and methods

### Cell lines and culture conditions

ATDC5 cells (mouse embryo teratocarcinoma-derived chondrogenic cell line) were purchased from European Collection of Cell Cultures (ECACC, Public Health England, Porton Down, UK), and were cultured at a seeding density of $4 \times 10^4$ cells/well for a 12-multiwell plate, $6 \times 10^4$ cells/well for a 6-multiwell plate and $7 \times 10^3$ cells/well for a 8-multiwell chamber slide, in a 1:1 mixture of Dulbecco's modified Eagle's and Ham's F12 medium (Flow Laboratories, Irvine, UK) supplemented with 5% fetal bovine serum (FBS: GIBCO, New York, NY, USA), 10 μg/ml bovine insulin (I; Wako Pure Chemical, Osaka, Japan), 10 μg/ml human transferrin (T; Boehringer Mannheim, Mannheim, Germany), and $3 \times 10^{-8}$ M sodium selenite (S; Sigma Chemical Co., St. Louis, MO, USA) at 37°C in a humidified atmosphere of 5% $CO_2$ in air for the initial 3 weeks, as previously described [10, 15]. On day 21, the culture medium was switched to alpha modified essential medium supplemented with 5% FBS plus ITS, and the $CO_2$ concentration was shifted to 3% to facilitate mineralization, as previously described [10, 15]. The medium was replaced every other day. To characterize the cells that are not subjected to chondrogenic media, ATDC5 cells were also cultured without ITS (S1 Fig). For the early stage experiment, IL-1β and/or sclerostin were filled in each well from 3 days to 3 weeks. For the late stage experiment, IL-1β and/or sclerostin were filled in each well from 3 weeks to 7 weeks. To mimic the OA-like condition, 10 ng/ml recombinant murine IL-1β (PeproTech, Rocky Hill, NJ, USA) was used, as previously described [12–14]. The effect of sclerostin was examined by the addition of 200 ng/ml recombinant mouse SOST (R&D systems).

### Alcian blue staining

To visualize the deposition of sulfated glycosaminoglycan (sGAG), a marker for chondrogenic differentiation, cells were fixed with 100% methanol and stained with 0.1% Alcian blue 8GS (Sigma) in 0.1 N HCl for 4 h at room temperature, as previously described [10].

## sGAG assay

The culture media were collected and sGAG content was quantified using a commercially available sGAG Alcian blue binding assay kit (Euro-Diagnostica, Malmo, Sweden). The absorbance at 640 nm was measured using a microplate reader (Infinite F50, TECAN, Kawasaki, Japan), as previously described [10].

## Alizarin red staining

To evaluate visualize the calcium deposits, cells were fixed with phosphate-buffered formalin and then stained with 40 mM alizarin red S (pH 4.2, Sigma) for 30 min, as previously described [10]. The Alizarin red-stained areas were scanned using an image scanner and analyzed qualitatively using Image J software.

## Immunostaining

The cultured cells were washed one time with cold PBS and fixed with 4% paraformaldehyde for 10 minutes at room temperature. Following washing three times with PBS, the cells were permeabilized with 0.2% Triton X-100 in PBS buffer for 20 minutes at room temperature. Following washing three times with 0.1% Tween20 in PBS, the cells were blocked with 0.1% Tween20 and 10% goat serum, with 1% BSA in PBS buffer for 1 hour at room temperature. The cells were incubated with the anti-rabbit primary antibodies of β-catenin (Abcam, Cambridge, UK), Axin1 (Novus Biologicals, Centennial, USA), Axin2 (Novus Biologicals) and phosphorylated LRP6 (Biorbyt Ltd., Cambridge, UK) over night at 4˚C. The cells were washed three times with 0.1% Tween20 in PBS. The cells were incubated with Alexa Fluor® 568 conjugated goat anti-rabbit IgG secondary antibody (Invitrogen, Carlsbad, USA) for 45 minutes at room temperature. To visualize the nuclei, the cells were double-stained with 4', 6-diamidino-2-phenylindole (DAPI) (Vector Laboratories, Burlingame, USA). The cells were viewed with a Keyence BZ 800 epifluorescence microscope, which was equipped with a digital camera (CFI 60, Nikon Corporation, Tokyo, Japan). All immunofluorescence images were obtained with identical exposure settings.

## Total RNA isolation and real-time RT-PCR

As described in detail previously [10], mRNA expression levels were analyzed following procedures. Total RNA was extracted from the cultured cells using Trizol reagent according to the manufacturer's instructions (Invitrogen, Carlsbad, CA, USA). RNA was quantified by measuring absorbance at 260 nm, and the quality was confirmed by 260/280 nm absorbance ratio greater than 1.8. First-strand cDNA synthesis from total RNA was performed using an iScriptTM advanced cDNA synthesis kit (BIO-RAD, Richmond, CA, USA). Quantitative real-time PCR was carried out using TaqMan gene expression assays (Applied Biosystems, Foster City, CA, USA) on a CFX96TM real-time PCR detection system (BIO-RAD). Expression of the gene of interest was normalized to GAPDH expression. TaqMan gene expression assays used in this study were as follows: *Col2a1* (Mm01309565_m1); *Col10a1* (Mm00487041_m1); *Sox9* (Mm00448840_m1); *Runx2* (Mm00501584_m1); *BMP2* (Mm01340178_m1); *Wnt3a* (Mm03053669_s1); *Wnt5a* (Mm00437347_m1); *LRP5* (Mm01227476_m1); *LRP6* (Mm00999795_m1); *Axin1* (Mm01299060_m1); *Axin2* (Mm00443610_m1); *Ctnnbl1* (β-catenin) (Mm00499427_m1); *MMP13* (Mm00439491_m1); *ADAMTS5* (Mm00478620_m1); and *GAPDH* (Mm99999915_g1).

## Statistical analysis

All experiments were repeated independently at least three times. All data are presented as means ± standard deviation. The analysis was performed with JMP Pro 12 software (SAS Institute Inc.) for Mac. Continuous variables were expressed as means. One-way analysis of variance (ANOVA) was used to compare mean values from different samples. Tukey's HSD was used for post-hoc analyses. A value of $P < 0.05$ was considered significant.

## Results

### No restorative effects of sclerostin on IL-1β-induced impairment in the early stage of chondrogenic differentiation

To simulate the inflammatory environment, 10 ng/ml IL-1β was added to the culture media for chondrogenic differentiation, and the effects of sclerostin were assessed. The mRNA expressions of markers for chondrogenic differentiation were significantly decreased with IL-1β treatment, and they were not restored by sclerostin (Fig 1a). The mRNA expression of MMP-13, one of markers for cartilage catabolism, was significantly increased with IL-1β treatment, and they were not significantly altered by sclerostin (Fig 1b). Chondrogenic differentiation with proteoglycan synthesis was confirmed by positive staining with Alcian blue after 3 weeks (Fig 1c). Less intense staining was observed with the addition of IL-1β, and it was not restored by sclerostin. The expression of Wnt/β-catenin-associated genes, *Wnt5a*, *LRP5*, *LRP6*, *Axin1*, *Axin2*, and *Ctnb-1*, was significantly decreased with IL-1β treatment, and was not restored by sclerostin (Fig 2a). Less intense expressions of Axin1, Axin2, and β-catenin was observed in the immunofluorescence images of the cultured cells with the addition of IL-1β, and these were not restored by sclerostin (Fig 2b–2d). These results suggested that IL-1β induces impairment in the early stage of chondrogenic differentiation and downregulation of the Wnt/β-catenin signaling. In this condition, there are no restorative effects of sclerostin on IL-1β-induced impairment in the early stage of chondrogenic differentiation.

### Inhibitory effects of sclerostin on IL-1β-induced terminal calcification in the late stage of chondrogenic differentiation

To assess the effects of sclerostin on IL-1β-induced terminal calcification in the late stage of chondrogenic differentiation, IL-1β and sclerostin were added from 3 weeks of culture. The mRNA expression of SOST was significantly decreased by IL-1β (S2 Fig). The mRNA expressions of markers for chondrogenic differentiation were significantly decreased with IL-1β, but they were restored by sclerostin (Fig 3a). The mRNA expressions of markers for cartilage catabolism and BMP2 were significantly increased with IL-1β, but they were restored by sclerostin (Fig 3b and 3c). Terminal calcification was confirmed by positive staining with Alizarin red after 7 weeks (Fig 3d). Stronger staining was identified with addition of IL-1β, but it was inhibited by sclerostin. The expression of Wnt/β-catenin associated genes, *Wnt3a*, *Wnt5a*, *LRP6*, *Axin1*, and *Ctnb-1*, was significantly increased with IL-1β, and was restored by sclerostin (Fig 4a). The immunofluorescence images of the cultured cells with the addition of IL-1β showed more intense expressions of phosphorylated LRP6, Axin1, Axin2, and β-catenin, but these were diminished by sclerostin (Fig 4b–4e). These results suggested that IL-1β promotes cartilage degradation and terminal calcification by upregulating the Wnt/β-catenin signaling pathway, and those effects were inhibited by sclerostin.

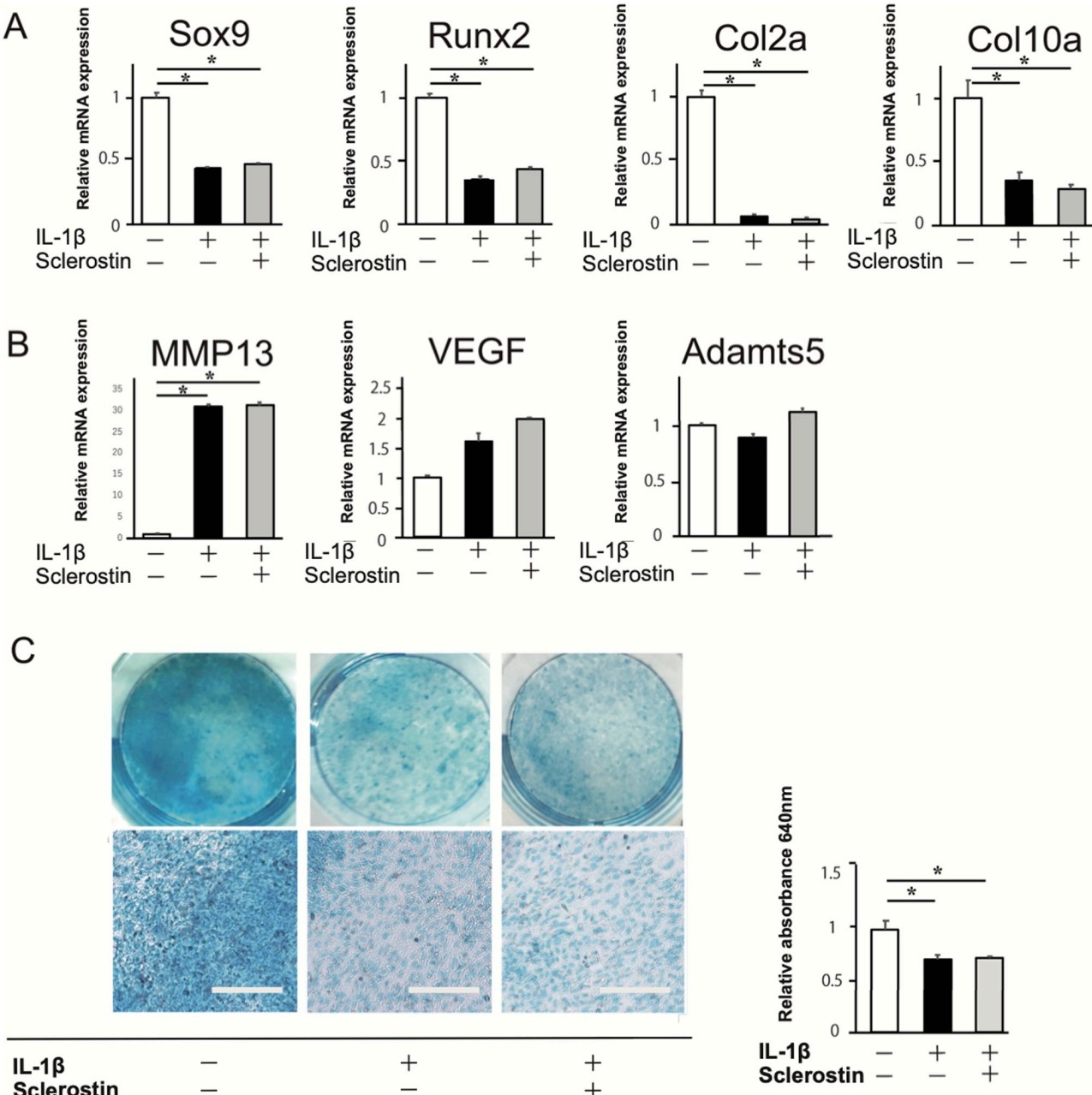

**Fig 1. Effect of sclerostin on early stage of chondrogenic differentiation in the presence of IL-1β.** ATDC5 cells were cultured for chondrogenic conditions, and 10 ng/ml IL-1β and/or 200 ng/ml sclerostin were added to the culture medium from 3 days to 3 weeks. (A) Relative mRNA expressions of markers for chondrogenic differentiations, *Sox9*, *Runx2*, *Col2a1*, and *Col10a*. (B) Relative mRNA expressions of markers for cartilage catabolism, *VEGF*, *MMP13*, and *Adamts5*. (C) Alcian Blue staining of ATDC5 cells under chondrogenic culture at 3 weeks of culture (left). Relative absorbance indicating GAG concentration of culture medium after 3 weeks (right). N = 4 *P<0.05.

## Discussion

The most important findings of this study were that sclerostin restores the chondrogenic phenotype and inhibits endochondral ossification under the IL-1β-induced condition. This process was associated with the downregulation of Wnt/β-catenin signaling. However, sclerostin did not alter the IL-1β-impaired early chondrogenesis from the undifferentiated state into

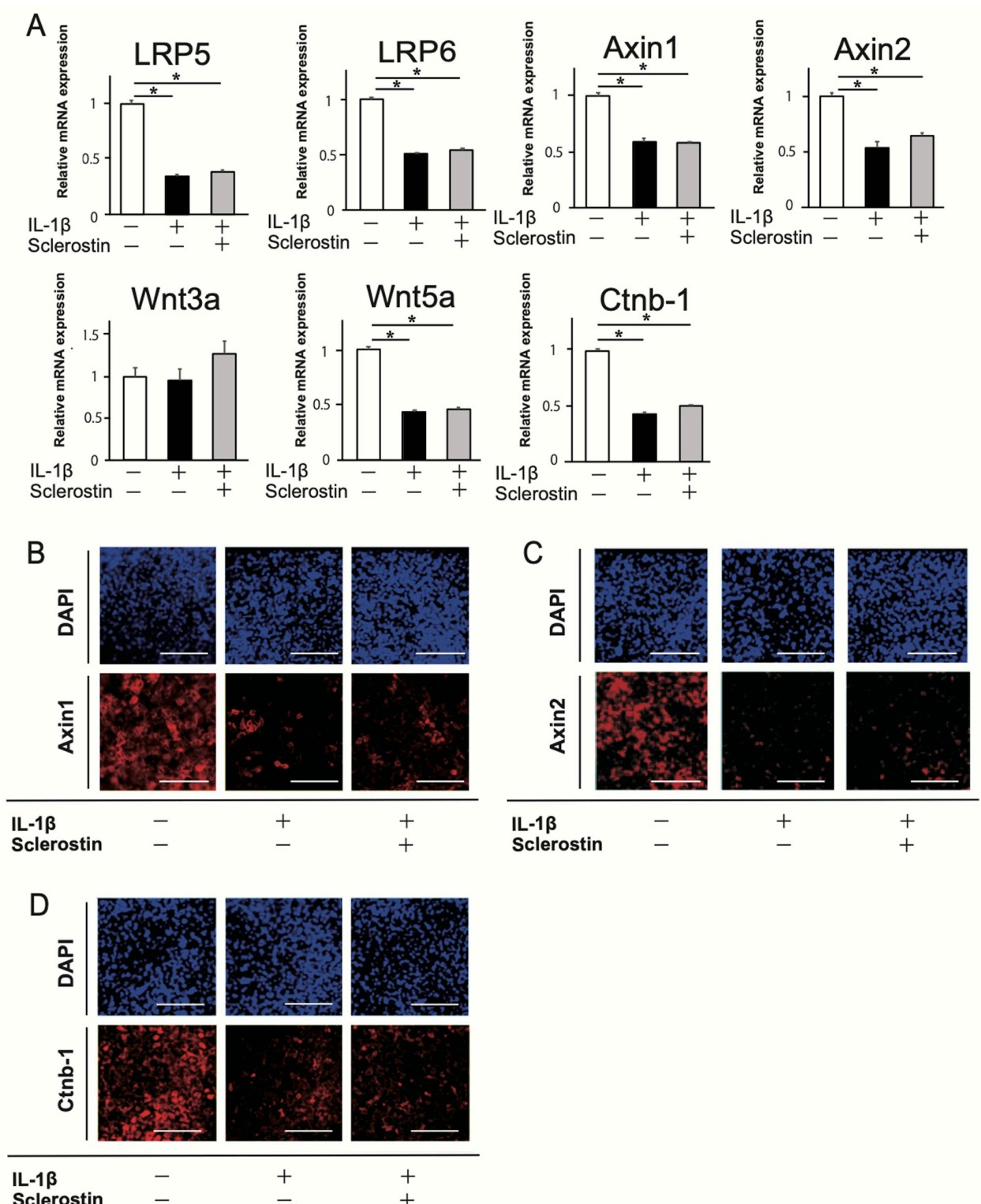

**Fig 2. Effect of sclerostin on Wnt/β-catenin signaling pathway in early stage of chondrogenic differentiation with IL-1β.** (A) Relative mRNA expressions of *LRP5*, *LRP6*, *Axin1*, *Axin2*, *Wnt3a*, *Wnt5a*, and *Ctnb-1*. N = 4 *P<0.05. (B-D) Immunofluorescence images of the cultured cells expressing Axin1 (B), Axin2 (C), and β-catenin (D). Scale bars = 100 μm.

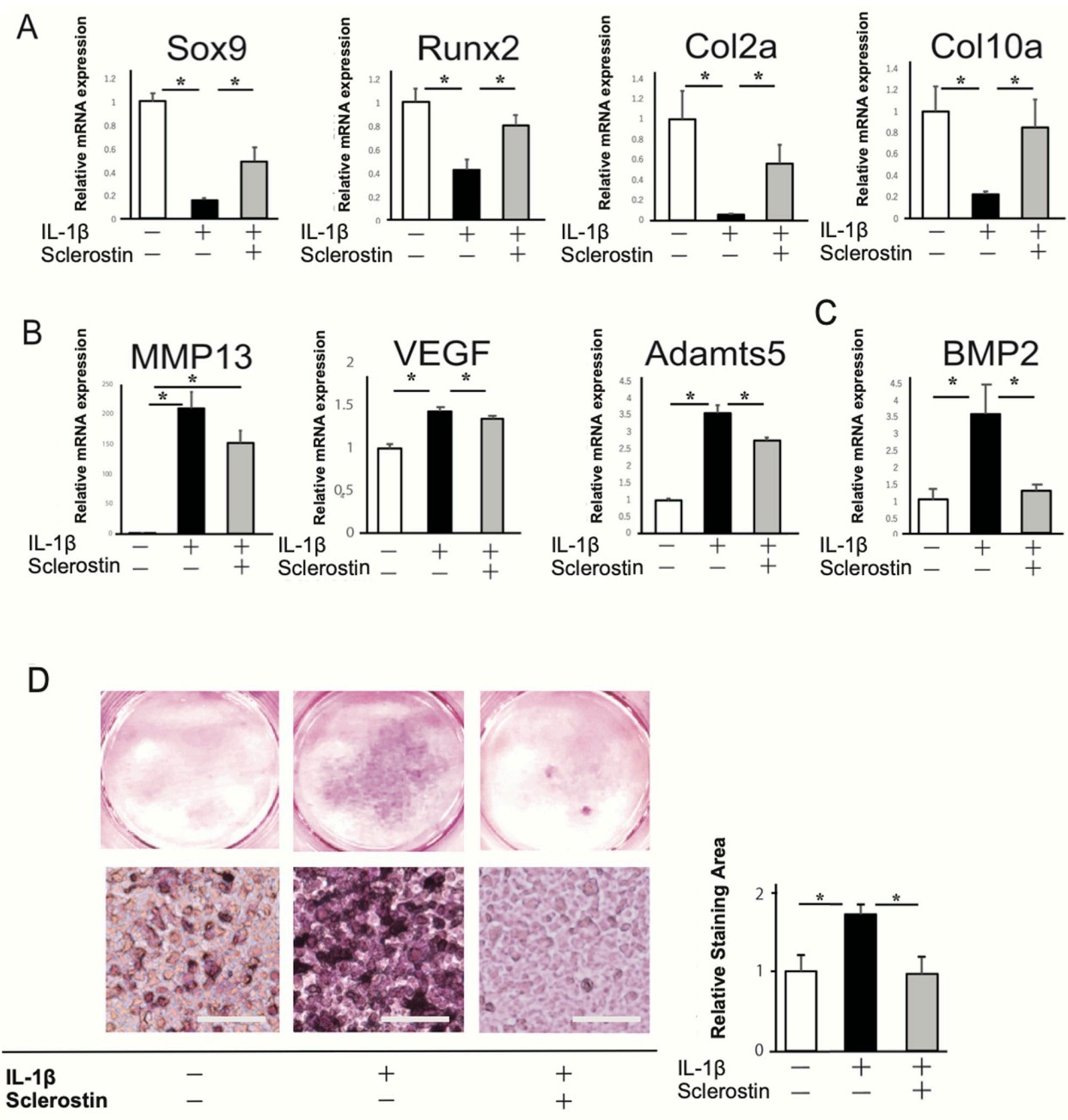

**Fig 3. Effect of sclerostin on late stage of chondrogenic differentiation in the presence of IL-1β.** ATDC5 cells were cultured for chondrogenic conditions, and 10 ng/ml IL-1β and/or 200 ng/ml sclerostin were added to the culture medium from 21 days. (A) Relative mRNA expressions of markers for chondrogenic differentiations, *Sox9*, *Runx2*, *Col2a1*, and *Col10a*. (B) Relative mRNA expressions of markers for cartilage catabolism, *VEGF*, *MMP13*, and *Adamts5*. (C) Relative mRNA expression of *BMP2*. (D) Alizarin red staining of ATDC5 cells under condition of endochondral ossification at 7 weeks of culture (left). Relative staining area indicating total size of calcified nodules (right). N = 4 *P<0.05.

proliferative chondrocytes. The major advance in the present study is the investigation of the restorative effects of sclerostin on chondrogenic differentiation in the multistep process from mesenchymal chondroprogenitor to terminal calcification *in vitro*, which can be separately assessed during the early and late stages of chondrogenic differentiation.

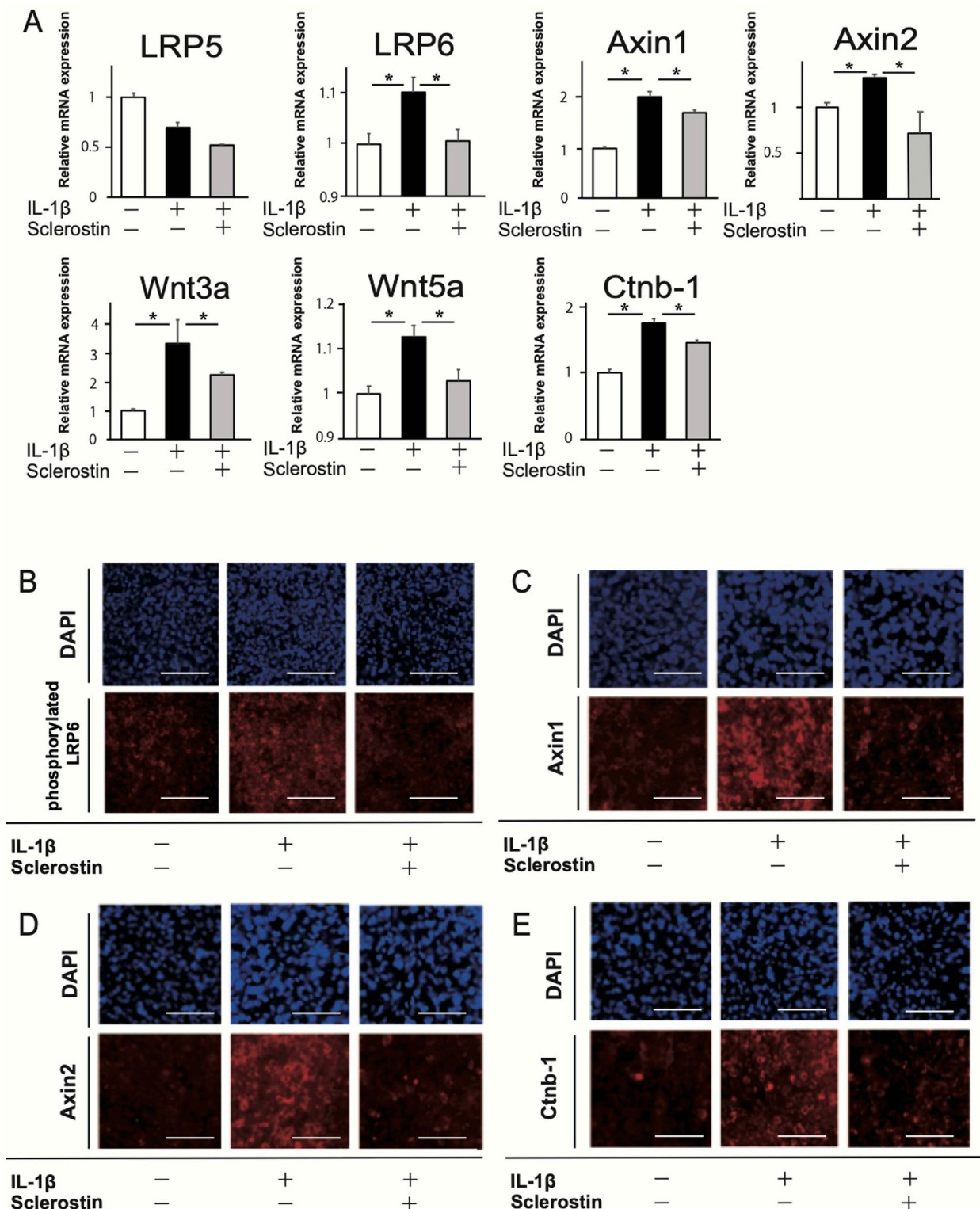

**Fig 4. Effect of sclerostin on Wnt/β catenin signaling pathway in late stage of chondrogenic differentiation with IL-1β.** (A) Relative mRNA expressions of *LRP5*, *LRP6*, *Axin1*, *Axin2*, *Wnt3a*, *Wnt5a*, and *Ctnb-1*. N = 4 *P<0.05. (B-E) Immunofluorescence images of the cultured cells expressing phosphorylated LRP6 (B), Axin1 (C), Axin2 (D), and β-catenin (E). Scale bars = 100 μm.

As described in detail previously [15, 16], ATDC5 is a good model system for studying the dynamic processes of chondrogenesis, and many findings in this system may have relevance to chondrogenesis *in vivo*. In our previous study, the role of sclerostin as an inhibitor of the canonical Wnt signaling pathway in the chondrogenic differentiation could be characterized using the same model system [11]. This established model system was used to investigate the effects on chondrogenic differentiation at the different timing of early and later stages in the present study. Although several in vivo OA models have been developed to investigate the pathological feature and therapeutic effects, they need to be considered influence under the multifactorial and complex conditions including mechanical load, synovial inflammation, cartilage degeneration and abnormal bone remodeling. This study simply focused the effect of sclerostin on the cytokine among several factors associated with OA pathogenesis.

The process of endochondral ossification, including chondrocyte hypertrophy, production of proteinases and cartilage apoptosis, is thought to be involved in the initiation and progression of OA [10, 17]. Wnt/β-catenin signaling plays a key role in the development of endochondral ossification and regulates OA development [3, 5]. As described in detail previously [3], β-catenin-dependent canonical Wnt signaling is required for the progression of endochondral ossification and growth of axial and appendicular skeletons, while excessive activation of this signaling can cause severe inhibition of initial cartilage formation and growth plate organization and function. Increased canonical Wnt signaling inhibits chondrogenesis [18, 19], but once cartilage has formed, it promotes chondrocyte maturation, enhances perichondral bone formation, initiates cartilage vascularization, and drives the formation of primary and secondary ossification centers [20]. Sclerostin is an inhibitor of Wnt/β-catenin signaling, which is expressed in the chondrocyte and modulates chondrogenic differentiation [11]. Thus, the present study focused on sclerostin as a potential target for the suppression of OA.

The present study investigated the restorative effects of sclerostin on chondrogenic differentiation under the IL-1β-induced condition in the different timing. IL-1β is a primary mediator of local inflammatory processes in OA [21, 22], and IL-1β-induced degradation of chondrogenesis is often utilized for *in vitro* model of OA [23, 24]. Since IL-1β is not cytotoxic up to a concentration of 100 ng/mL [12], and 10 ng/mL IL-1β is considered as an optimized concentration to induce OA-like condition *in vitro*, 10 ng/mL of concentration was used in this study [13, 14]. IL-1β modulates the chondrogenic differentiation, and those effects differ between the early and late stages. In the early stage, IL-1β downregulated Wnt/β-catenin signaling and impaired chondrogenic differentiation. In this condition, Wnt/β-catenin signaling has already been inhibited, and there was no further requirement for sclerostin to act as Wnt inhibitor. In addition, IL-1β downregulates the synthesis of master chondrogenic factor Sox9 [25], and inhibits early differentiation from mesenchymal phenotype into proliferative chondrocytes [26, 27]. In contrast, IL-1β promotes endochondral ossification with increased expressions of catabolic markers in the later stage [28, 29]. Although details of IL-1β and Wnt cross talk have not been well elucidated, as a possible mechanism, it was reported that nitric oxide mediates the IL-1β-induced inflammatory response of chondrocytes through the upregulation of Wnt signaling [30]. In the present study, sclerostin inhibited the IL-1β-induced terminal calcification through downregulation of Wnt/β-catenin signaling. Thus, the restorative effects of sclerostin on chondrogenic phenotype under the IL-1β-induced condition may be expected only in the later stage of chondrogenic differentiation.

In summary, the present study demonstrated that sclerostin restores the chondrogenic phenotype and inhibits endochondral ossification under the IL-1β-induced OA-like environment. IL-1β reduces the production of the Wnt antagonist, and enhances Wnt signaling in the articular resident cells [31]. Suppression of the Wnt signaling by the antagonist may play a key role

in the maintenance of articular homeostasis [9, 32], and has the potential to prevent the progression of OA. Thus, sclerostin is a candidate treatment option for OA.

## Supporting information

**S1 Fig. To characterize the cells that are not subjected to chondrogenic media, ATDC5 cells were cultured with or without ITS.** (A) Alcian Blue staining of ATDC5 cells at 3 weeks of culture in chondrogenic media (left) and non-chondrogenic media (right). Less intense staining is observed in non-chondrogenic media. Scale bars = 100 μm. (B) Alizarin red staining of ATDC5 cells at 7 weeks of culture in chondrogenic media (left) and non-chondrogenic media (right). Less intense staining is observed in non-chondrogenic media. Scale bars = 100 μm.
(TIFF)

**S2 Fig. To assess the effect of IL-1ß on expression of sclerostin, ATDC5 cells were cultured for 3 weeks in chondrogenic media and then 10 ng/ml IL-1β was added to the media for 24 hours.** The relative mRNA expression of SOST is significantly decreased in the cells with addition of IL-1β. N = 4 $^*$P<0.05.
(TIFF)

## Author Contributions

**Conceptualization:** Ken Kumagai.

**Data curation:** Kazuma Miyatake, Yasuteru Yamaguchi.

**Formal analysis:** Kazuma Miyatake, Ken Kumagai, Yasuteru Yamaguchi.

**Funding acquisition:** Ken Kumagai, Yutaka Inaba.

**Investigation:** Kazuma Miyatake, Ken Kumagai, Sosuke Imai, Yasuteru Yamaguchi.

**Methodology:** Kazuma Miyatake, Ken Kumagai, Yasuteru Yamaguchi.

**Supervision:** Ken Kumagai.

**Writing – original draft:** Kazuma Miyatake, Ken Kumagai.

**Writing – review & editing:** Sosuke Imai, Yasuteru Yamaguchi, Yutaka Inaba.

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
