## [Decision Letter · Decision Letter 0]

4 Feb 2020

PONE-D-19-33198

Sclerostin inhibits interleukin-1β-induced late stage chondrogenic differentiation through downregulation of Wnt/β-catenin signaling pathway

PLOS ONE

Dear Dr Kumagai,

Thank you for submitting your manuscript to PLOS ONE. After careful consideration, we feel that it has merit but does not fully meet PLOS ONE’s publication criteria as it currently stands. Therefore, we invite you to submit a revised version of the manuscript that comprehensively addresses the points raised by the three reviewers. Note that the reviewers have made a number of highly pertinent comments, each one of which needs to be dealt with in the revised manuscript.

We would appreciate receiving your revised manuscript by Mar 20 2020 11:59PM. To enhance the reproducibility of your results, we recommend that if applicable you deposit your laboratory protocols in protocols.io, where a protocol can be assigned its own identifier (DOI) such that it can be cited independently in the future. For instructions see: http://journals.plos.org/plosone/s/submission-guidelines#loc-laboratory-protocols

We look forward to receiving your revised manuscript.

Kind regards,

Michael Schubert

Academic Editor

PLOS ONE

Journal Requirements:

https://journals.plos.org/plosone/article?id=10.1371%2Fjournal.pone.0201839

We appreciate that the above is your own previously published work, but do not feel that text reuse outside of the Material and Methods is best practice and we would ask that you rephrase these sections. In addition as part of your revision we would recommend that you clearly indicate where methods are based on methods previously described in previous publication (e.g. As previously described [Reference],...)

Further consideration is dependent on these concerns being addressed.

"This work was supported in part by a Grant-in-Aid for Scientific Research from

the Ministry of Education, Culture, Sports, Science and Technology of Japan (#25462347)."

The authors received no specific funding for this work.

Reviewers' comments:

Reviewer's Responses to Questions

**Comments to the Author**

1. Is the manuscript technically sound, and do the data support the conclusions?

Reviewer #1: Partly

Reviewer #2: Partly

Reviewer #3: Partly

2. Has the statistical analysis been performed appropriately and rigorously? 

Reviewer #1: I Don't Know

Reviewer #2: Yes

Reviewer #3: Yes

3. Have the authors made all data underlying the findings in their manuscript fully available?

Reviewer #1: Yes

Reviewer #2: No

Reviewer #3: No

4. Is the manuscript presented in an intelligible fashion and written in standard English?

Reviewer #1: Yes

Reviewer #2: Yes

Reviewer #3: Yes

5. Review Comments to the Author

Reviewer #1: In this manuscript, the authors use the ATDC5 cell line to examine the dual effects of sclerostin and interleukin 1 beta on cell differentiation, cell mineralisation and gene expression of chondrocyte markers and components of the wnt signalling pathway. They found that IL-1B impaired chondrocyte differentiation, and that the addition of sclerostin had no additional effect on this. IL-1B also drove chondrocyte mineralisation and sclerostin was able to restore this.

1) I am unclear as to the novelty of this work, as the dual effects of sclerostin and inflammatory cyotkines have been previously investigated (e.g. doi: 10.1016/j.joca.2011.04.014; doi: 10.3892/mmr.2017.6278; doi: 10.1186/s13075-015-0540-6).

2) I have concerns with some of the data - in particular the alizarin red staining (and alcian blue) which appear to have staining outside of the well and therefore make me question these results. Higher power is also needed to confirm this is not ectopic calcification. Further, some of the stats are questionable - e.g. Fig 3B VEGF has huge error bars which are significant?

3) Some of the statements in the results are incorrect e.g. line 118 - only one marker was significantly changed and this statement suggests all were. This needs rectifying throughout

4) Were other markers of endochondral ossificaion examined - e.g. PTH/PTHrP/Ihh? The authors looked at ADAMTS5, but what about TIMPs?

5) Justification of concentrations used required

Reviewer #2: In this work, Miyatake and colleagues provide a straightforward study showing the potential of sclerostin to inhibit canonical Wnt signaling and be a therapeutic target for osteoarthritis (OA). This study provides incremental knowledge of sclerostin’s role in an IL1-β induced OA in vitro model. I am not convinced that the cell line used in this study is the best model to look at early and later stages of chondrogenic differentiation in the context of endochondral ossification, especially as these are teratoma-derived if I understand correctly. The authors need to speak to why this cell line was used. Below are additional comments to the authors:

Introduction

- Would be beneficial to reference what is known from in vivo models of sclerostin and canonical Wnt pathway components, specifically knock-out or transgenic mouse models, to show physiological, whole system effects.

Methods

- Need to provide more detail about the source/company, species, and phenotype of the ATDC5 cells.

- Concerned about cells cultured through 7 weeks in 2D and the environment affecting phenotype more than the pharmacological treatment

- Line 63, clarify that for early stage treatment was started at 24 hours after confluency and for late stage, treatment was started on day 21 as the wording right in this current draft is confusing.

- Why standard error of the mean vs standard deviation for plots?

- Need to explain and/or reference why 10 ng/mL IL-1β was used to induce inflammation vs. other concentrations.

- Need a control group of cells that have not been subjected to chondrogenic media to characterize and compare how these cells behave over the treatment period in 2D culture.

- Would be helpful to look at protein levels (immunostaining if not westerns) of active beta-catenin and downstream targets (e.g. Axin 1/2) to further confirm sclerostin’s effects.

Discussion

- Would be beneficial to provide commentary on how this knowledge could be used for treatment in diseases such as osteoarthritis, in context with this in vitro model compared to in vivo models.

Reviewer #3: This manuscript reports that sclerostin prevents late stage of chondrogenic differentiation induced by interleukin-1ß in ATDC5 cells through inhibition of Wnt/ß-catenin signaling. The authors show that sclerostin has no effect on the expression of chondrogenic differentiation markers and Wnt/ß-catenin pathway genes in response to interleukin-1ß at early stage (Figs 1 and 2). However, sclerostin counteracts the effect of interleukin-1ß on the expression of most these genes at late stage (Figs 3 and 4). Although this study indicates an antagonistic effect of sclerostin on interleukin-1ß in chondrogenic differentiation, the present results are too preliminary to warrant a publication.

1. Interleukin-1ß treatment increases the expression of some Wnt/ß-catenin genes, and inhibits the expression of others (for example, axin2), but the extent of increase or decrease is very limited for most genes. What would be the overall outcome of these regulations on Wnt/ß-catenin signaling? It is necessary to assess whether Wnt/signaling is indeed increased in the cells treated interleukin-1ß. Similarly, the authors should also examine whether sclerostin exerts a net effect on Wnt/signaling in cells treated by interleukin-1ß.

2. It is not clear how sclerostin regulates the expression of these genes in the absence of interleukin-1ß. It inhibits the expression of Wnt/ß-catenin pathway genes directly or indirectly by inhibiting bone formation? Sclerostin is a secreted protein that antagonizes Wnt signaling by binding to LRP5/6, how it inhibits axin1 and ctnb-1 (ß-catenin) expression, is there a positive feedback of Wnt/ß-catenin signaling in the cells at this stage of chondrogenic differentiation.

3. The effect of sclerostin on BMP signaling needs to be examined to see if it inhibits chondrogenic differentiation also by antagonizing this pathway.

6. PLOS authors have the option to publish the peer review history of their article (what does this mean?). If published, this will include your full peer review and any attached files.

Reviewer #1: No

Reviewer #2: No

Reviewer #3: No

---

## [Author Response · Author response to Decision Letter 0]

24 Jul 2020

Please find in the following paragraphs a point-by-point reply to the comments of the reviewers.

Reviewer #1: In this manuscript, the authors use the ATDC5 cell line to examine the dual effects of sclerostin and interleukin 1 beta on cell differentiation, cell mineralisation and gene expression of chondrocyte markers and components of the wnt signalling pathway. They found that IL-1B impaired chondrocyte differentiation, and that the addition of sclerostin had no additional effect on this. IL-1B also drove chondrocyte mineralisation and sclerostin was able to restore this.

1) I am unclear as to the novelty of this work, as the dual effects of sclerostin and inflammatory cyotkines have been previously investigated (e.g. doi: 10.1016/j.joca.2011.04.014; doi: 10.3892/mmr.2017.6278; doi: 10.1186/s13075-015-0540-6).

Response: The strong point of this study is the investigation of these effects through whole process of chondrogenic differentiation from mesenchymal chondroprogenitor to terminal calcification. Especially, the role of inflammatory cyotkines and sclerostin in the terminal calcification of chondrocyte has not been investigated in the previous studies, which it is considered as the novelty of the present study. Additional comments were described in the discussion (Line 186-198).

2) I have concerns with some of the data - in particular the alizarin red staining (and alcian blue) which appear to have staining outside of the well and therefore make me question these results. Higher power is also needed to confirm this is not ectopic calcification. Further, some of the stats are questionable - e.g. Fig 3B VEGF has huge error bars which are significant?

Response: The images of alizarin red and alcian blue were replaced with new ones. Photomicrographs with high power magnification were added to the data. Relative mRNA expression of VEGF was re-analyzed by real-time PCR, and the data were replaced in Fig. 3B.

3) Some of the statements in the results are incorrect e.g. line 118 - only one marker was significantly changed and this statement suggests all were. This needs rectifying throughout

Response: We apologize this incorrect writing. The description ‘The mRNA expressions of markers for cartilage catabolism were’ was revised to ‘The mRNA expression of MMP-13, one of markers for cartilage catabolism, was’ (Line 142-143).

4) Were other markers of endochondral ossificaion examined - e.g. PTH/PTHrP/Ihh? The authors looked at ADAMTS5, but what about TIMPs?

Response: Unfortunately, the other markers including PTH/PTHrP/Ihh and TIMPs were not investigated. We would like to analyze them in the future opportunity.

5) Justification of concentrations used required

Response: IL-1β was not cytotoxic up to a concentration of 100 ng/mL (Graeser et al. 2009). Of these concentrations, 10 ng/mL IL-1β is considered as an optimized concentration to induce OA-like condition in vitro, and this concentration was also used in most of previous studies (Shi et al. 2016, Wang et al. 2019). Text was revised (Line 217- 219) and following references were added.

12. Graeser AC, et al. Synergistic chondroprotective effect of alpha-tocopherol, ascorbic acid, and selenium as well as glucosamine and chondroitin on oxidant induced cell death and inhibition of matrix metalloproteinase-3--studies in cultured chondrocytes. Molecules. 2009; 15: 27-39.

13. Shi S, et al. Silencing of Wnt5a prevents interleukin-1beta-induced collagen type II degradation in rat chondrocytes. Exp Ther Med. 2016; 12: 3161-3166.

14. Wang F, et al. IL-1beta receptor antagonist (IL-1Ra) combined with autophagy inducer (TAT-Beclin1) is an effective alternative for attenuating extracellular matrix degradation in rat and human osteoarthritis chondrocytes. Arthritis Res Ther. 2019; 21: 171.

Reviewer #2: In this work, Miyatake and colleagues provide a straightforward study showing the potential of sclerostin to inhibit canonical Wnt signaling and be a therapeutic target for osteoarthritis (OA). This study provides incremental knowledge of sclerostin’s role in an IL1-β induced OA in vitro model. I am not convinced that the cell line used in this study is the best model to look at early and later stages of chondrogenic differentiation in the context of endochondral ossification, especially as these are teratoma-derived if I understand correctly. The authors need to speak to why this cell line was used.

Response: The following description was added to the revised manuscript (Line 186-198).

As described in detail previously [15, 16], ATDC5 is a good model system for studying the dynamic processes of chondrogenesis, and many findings in this system may have relevance to chondrogenesis in vivo. In our previous study, the role of sclerostin as an inhibitor of the canonical Wnt signaling pathway in the chondrogenic differentiation could be characterized using the same model system [11]. This established model system was used to investigate the effects on chondrogenic differentiation at the different timing of early and later stages in the present study. Although several in vivo OA models have been developed to investigate the pathological feature and therapeutic effects, they need to be considered influence under the multifactorial and complex conditions including mechanical load, synovial inflammation, cartilage degeneration and abnormal bone remodeling. This study simply focused the effect of sclerostin on the cytokine among several factors associated with OA pathogenesis.

Below are additional comments to the authors:

Introduction

- Would be beneficial to reference what is known from in vivo models of sclerostin and canonical Wnt pathway components, specifically knock-out or transgenic mouse models, to show physiological, whole system effects.

Response: Descriptions about in vivo models of sclerostin and canonical Wnt pathway components were added to the introduction section (Line 33-35, 40-43).

Methods

- Need to provide more detail about the source/company, species, and phenotype of the ATDC5 cells.

Response: Accordingly, description about ATDC5 was added to the text (Line 58-60).

- Concerned about cells cultured through 7 weeks in 2D and the environment affecting phenotype more than the pharmacological treatment

Response: Since it is difficult to create the model for terminal calcification in 3D culture of chondrocyte, we used 2D culture model. Addition of IL-1β accelerated this phenotypic change, but treatment with sclerostin showed significant effects of restoration even in this environment.

- Line 63, clarify that for early stage treatment was started at 24 hours after confluency and for late stage, treatment was started on day 21 as the wording right in this current draft is confusing.

Response: The description was revised as suggested (Line 73-75).

- Why standard error of the mean vs standard deviation for plots?

Response: It was our mistake. All data were given as mean and standard deviation. The statement was revised (Line 130).

- Need to explain and/or reference why 10 ng/mL IL-1β was used to induce inflammation vs. other concentrations.

Response: IL-1β was not cytotoxic up to a concentration of 100 ng/mL (Graeser et al. 2009). Of these concentrations, 10 ng/mL IL-1β is considered as an optimized concentration to induce OA-like condition in vitro, and this concentration was also used in most of previous studies (Shi et al. 2016, Wang et al. 2019). Text was revised and following references were added (Line 217- 219).

12. Graeser AC, et al. Synergistic chondroprotective effect of alpha-tocopherol, ascorbic acid, and selenium as well as glucosamine and chondroitin on oxidant induced cell death and inhibition of matrix metalloproteinase-3--studies in cultured chondrocytes. Molecules. 2009; 15: 27-39.

13. Shi S, et al. Silencing of Wnt5a prevents interleukin-1beta-induced collagen type II degradation in rat chondrocytes. Exp Ther Med. 2016; 12: 3161-3166.

14. Wang F, et al. IL-1beta receptor antagonist (IL-1Ra) combined with autophagy inducer (TAT-Beclin1) is an effective alternative for attenuating extracellular matrix degradation in rat and human osteoarthritis chondrocytes. Arthritis Res Ther. 2019; 21: 171.

- Need a control group of cells that have not been subjected to chondrogenic media to characterize and compare how these cells behave over the treatment period in 2D culture.

Response: To characterize the cells that are not subjected to chondrogenic media, ATDC5 cells were also cultured without ITS. Chondrogenic differentiation with proteoglycan synthesis was confirmed by positive staining with Alcian blue after 3 weeks (Supple Fig. 1a). Less intense staining was observed without ITS. Terminal calcification was confirmed by positive staining with Alizarin red after 7 weeks (Supple Fig. 1b). Less staining was identified without ITS.

- Would be helpful to look at protein levels (immunostaining if not westerns) of active beta-catenin and downstream targets (e.g. Axin 1/2) to further confirm sclerostin’s effects.

Response: Accordingly, immunostaining for ß-catenin, Axin1, and Axin 2 was performed and data were added (Fig. 2 and Fig. 4).

Discussion

- Would be beneficial to provide commentary on how this knowledge could be used for treatment in diseases such as osteoarthritis, in context with this in vitro model compared to in vivo models.

Response: Following comment was added to the revised text (Line 193-198).

Although several in vivo OA models have been developed to investigate the pathological feature and therapeutic effects, they need to be considered influence under the multifactorial and complex conditions including mechanical load, synovial inflammation, cartilage degeneration and abnormal bone remodeling. This study simply focused the effect of sclerostin on the cytokine among several factors associated with OA pathogenesis.

Reviewer #3: This manuscript reports that sclerostin prevents late stage of chondrogenic differentiation induced by interleukin-1ß in ATDC5 cells through inhibition of Wnt/ß-catenin signaling. The authors show that sclerostin has no effect on the expression of chondrogenic differentiation markers and Wnt/ß-catenin pathway genes in response to interleukin-1ß at early stage (Figs 1 and 2). However, sclerostin counteracts the effect of interleukin-1ß on the expression of most these genes at late stage (Figs 3 and 4). Although this study indicates an antagonistic effect of sclerostin on interleukin-1ß in chondrogenic differentiation, the present results are too preliminary to warrant a publication.

1. Interleukin-1ß treatment increases the expression of some Wnt/ß-catenin genes, and inhibits the expression of others (for example, axin2), but the extent of increase or decrease is very limited for most genes. What would be the overall outcome of these regulations on Wnt/ß-catenin signaling? It is necessary to assess whether Wnt/signaling is indeed increased in the cells treated interleukin-1ß. Similarly, the authors should also examine whether sclerostin exerts a net effect on Wnt/signaling in cells treated by interleukin-1ß.

Response: As reviewer suggested, effects of interleukin-1ß on LRP5/6 seems to be limited. However, expression of phosphorylated LRP6 was increased by interleukin-1ß (Fig. 4b). Gene expression of Axin2 was re-examined and data showed the same trend as the other related genes. We apologize the data in the initial submission were not correct due to technical error and the graph was replaced in the revised figure. In addition, we assessed the expression of SOST in the cells during chondeogenic differentiation with or without Interleukin-1ß (Supple Fig. 2). Interleukin-1ß significantly suppressed the expression of SOST. This is thought to affect indirectly upregulation of Wn/ß-catenin signaling. Similar mechanism was reported by Yoshida et al (Inflammation 2018, [31]). Interleukin-1ß activated Wnt/ß-catenin signaling by suppression of Wnt antagonist, DKK-1.

2. It is not clear how sclerostin regulates the expression of these genes in the absence of interleukin-1ß. It inhibits the expression of Wnt/ß-catenin pathway genes directly or indirectly by inhibiting bone formation? Sclerostin is a secreted protein that antagonizes Wnt signaling by binding to LRP5/6, how it inhibits axin1 and ctnb-1 (ß-catenin) expression, is there a positive feedback of Wnt/ß-catenin signaling in the cells at this stage of chondrogenic differentiation.

Response: Our previous study (Yamaguchi et al. PLOS ONE 2018 [10]) demonstrated that sclerostin directly inhibited the expression of Wnt/ß-catenin pathway genes with downregulation of LRP5/6 during chondrogenic differentiation in the absence of interleukin-1ß. When added to the interleukin-1ß, sclerostin suppressed the expressions of LRP6 and phosphorylated LRP6 in the present study. It is unclear whether there is a positive feedback of Wnt/ß-catenin signaling in the cells.

3. The effect of sclerostin on BMP signaling needs to be examined to see if it inhibits chondrogenic differentiation also by antagonizing this pathway.

Response: The mRNA expression level of BMP-2 was additionally investigated and added to ta data (Fig 3c). BMP2 was upregulated by interleukin-1ß, and it was inhibited by sclerostin.

---

## [Decision Letter · Decision Letter 1]

12 Aug 2020

PONE-D-19-33198R1

Sclerostin inhibits interleukin-1β-induced late stage chondrogenic differentiation through downregulation of Wnt/β-catenin signaling pathway

PLOS ONE

Dear Dr. Kumagai,

Thank you for submitting your manuscript to PLOS ONE. After careful consideration, we feel that it has merit but does not fully meet PLOS ONE’s publication criteria as it currently stands. Therefore, we invite you to submit a revised version of the manuscript that addresses the points raised during the review process.

We look forward to receiving your revised manuscript.

Kind regards,

Michael Schubert

Academic Editor

PLOS ONE

Reviewers' comments:

Reviewer's Responses to Questions

**Comments to the Author**

1. If the authors have adequately addressed your comments raised in a previous round of review and you feel that this manuscript is now acceptable for publication, you may indicate that here to bypass the “Comments to the Author” section, enter your conflict of interest statement in the “Confidential to Editor” section, and submit your "Accept" recommendation.

Reviewer #1: All comments have been addressed

Reviewer #2: (No Response)

Reviewer #3: (No Response)

2. Is the manuscript technically sound, and do the data support the conclusions?

Reviewer #1: Yes

Reviewer #2: Yes

Reviewer #3: Partly

3. Has the statistical analysis been performed appropriately and rigorously? 

Reviewer #1: Yes

Reviewer #2: I Don't Know

Reviewer #3: Yes

4. Have the authors made all data underlying the findings in their manuscript fully available?

Reviewer #1: Yes

Reviewer #2: Yes

Reviewer #3: Yes

5. Is the manuscript presented in an intelligible fashion and written in standard English?

Reviewer #1: Yes

Reviewer #2: Yes

Reviewer #3: Yes

6. Review Comments to the Author

Reviewer #1: (No Response)

Reviewer #2: The authors have addressed all of the original comments and critiques except the issue with standard deviation vs. standard error of the mean. In the comments to reviewer, the authors comment that standard deviation was calculated and used to show variance in the data sets. However, the text reads "standard error" which is not standard deviation - as it is the standard deviation divided by the square root of the sample size. Please correct accurately and update in the text.

Reviewer #3: This revised manuscript includes additional experiments examining the effects of IL-1ß and sclerostin on the expression of Wnt/ß-catenin signaling components at early and late stages of chondrogenic differentiation in ATDC5 cells. It appears that IL-1ß inhibits the expression of both positive and negative regulators of Wnt/ß-catenin signaling at early stages of chondrogenic differentiation, and promotes their expression in terminal calcification at late stages of chondrogenic differentiation. In addition, sclerostin counteracts the effects of IL-1ß at late stages but not at early stages. However, the present version failed to address the issues raised in my previous review. In particular, it remains unclear whether IL-1ß inhibits Wnt/ß-catenin signaling at early stages of chondrogenic differentiation, and promotes Wnt/ß-catenin signaling at late stages of chondrogenic differentiation.

I understand that IL-1ß may exert differential effects at early and late stages of chondrogenic differentiation. Nevertheless, IL-1ß seems to regulate the expression of both positive and negative regulators of Wnt/ß-catenin signaling. Thus, the net outcome on Wnt/ß-catenin signaling is unclear.

The effect of sclerostin on IL-1ß at late stages of chondrogenic differentiation is likely indirect. Sclerostin binds to LRP5/6 to inhibit Wnt/ß-catenin signaling, and it is believed that the antagonistic effect of sclerostin on bone formation is mediated by Wnt signaling. Thus, there is no sufficient novelty in the present study.

It is difficult to understand how the experiments were done, in particular, when the early effects were analyzed after treatment. For example, the results presented in Fig. 1C seem to be obtained after 3 weeks, but it is not clear when the cells were treated, and they represent early or late effect?

7. PLOS authors have the option to publish the peer review history of their article (what does this mean?). If published, this will include your full peer review and any attached files.

Reviewer #1: No

Reviewer #2: No

Reviewer #3: No

---

## [Author Response · Author response to Decision Letter 1]

10 Sep 2020

Please find in the following paragraphs a point-by-point reply to the comments of the reviewers.

Reviewer #1: (No Response)

Reviewer #2: The authors have addressed all of the original comments and critiques except the issue with standard deviation vs. standard error of the mean. In the comments to reviewer, the authors comment that standard deviation was calculated and used to show variance in the data sets. However, the text reads "standard error" which is not standard deviation - as it is the standard deviation divided by the square root of the sample size. Please correct accurately and update in the text.

Response: We apologize incorrect description. In the text, "standard error" was corrected to "standard deviation" (Line 130).

Reviewer #3: This revised manuscript includes additional experiments examining the effects of IL-1ß and sclerostin on the expression of Wnt/ß-catenin signaling components at early and late stages of chondrogenic differentiation in ATDC5 cells. It appears that IL-1ß inhibits the expression of both positive and negative regulators of Wnt/ß-catenin signaling at early stages of chondrogenic differentiation, and promotes their expression in terminal calcification at late stages of chondrogenic differentiation. In addition, sclerostin counteracts the effects of IL-1ß at late stages but not at early stages. However, the present version failed to address the issues raised in my previous review. In particular, it remains unclear whether IL-1ß inhibits Wnt/ß-catenin signaling at early stages of chondrogenic differentiation, and promotes Wnt/ß-catenin signaling at late stages of chondrogenic differentiation.

I understand that IL-1ß may exert differential effects at early and late stages of chondrogenic differentiation. Nevertheless, IL-1ß seems to regulate the expression of both positive and negative regulators of Wnt/ß-catenin signaling. Thus, the net outcome on Wnt/ß-catenin signaling is unclear.

Response: IL-1β modulates the chondrogenic differentiation and Wnt/ß-catenin signaling, but the effects differ between the early and late stages. Previous studies reported that IL-1β impaired initial chondrogenic differentiation from mesenchymal cells [1, 2]. The mechanisms may be associated with downregulation of wnt/β-catenin signaling due to negative feedback by the interaction between Wnt/β-catenin signaling and the others such as NF-κB [3], although details are unclear. When the cells were treated with IL-1β from the beginning in the present in vitro culture system of chondrogenic differentiation, initial chondrogenic differentiation was inhibited and wnt/β-catenin signaling was downregulated in the early stage. But once chondrogenic differentiation was progressed, IL-1β upregulated wnt/β-catenin signaling in the late stage of chondrogenic differentiation and promoted endochondral ossification. This mechanism is supported by the other studies [4, 5].

References

1. Simsa-Maziel S, Monsonego-Ornan E. Interleukin-1beta promotes proliferation and inhibits differentiation of chondrocytes through a mechanism involving down-regulation of FGFR-3 and p21. Endocrinology. 2012; 153: 2296-310.

2. Murakami S, Lefebvre V, de Crombrugghe B. Potent inhibition of the master chondrogenic factor Sox9 gene by interleukin-1 and tumor necrosis factor-alpha. J Biol Chem. 2000; 275: 3687-92.

3. Ma B, van Blitterswijk CA, Karperien M. A Wnt/β-catenin negative feedback loop inhibits interleukin-1-induced matrix metalloproteinase expression in human articular chondrocytes. Arthritis Rheum. 2012; 64:2589-2600.

4. Mumme M, Scotti C, Papadimitropoulos A, Todorov A, Hoffmann W, Bocelli-Tyndall C, et al. Interleukin-1beta modulates endochondral ossification by human adult bone marrow stromal cells. Eur Cell Mater. 2012; 24: 224-36.

5. Scotti C, Piccinini E, Takizawa H, Todorov A, Bourgine P, Papadimitropoulos A, et al. Engineering of a functional bone organ through endochondral ossification. Proc Natl Acad Sci U S A. 2013; 110: 3997-4002.

The effect of sclerostin on IL-1ß at late stages of chondrogenic differentiation is likely indirect. Sclerostin binds to LRP5/6 to inhibit Wnt/ß-catenin signaling, and it is believed that the antagonistic effect of sclerostin on bone formation is mediated by Wnt signaling. Thus, there is no sufficient novelty in the present study.

Response: IL-1β inhibits the expression of sclerostin/SOST in ATDC5 cells (Supple Fig. 2). As a possible mechanism, the stimulatory effects of IL-1β on Wnt/β-catenin signaling may be mediated by regulating interaction between sclerostin and LRP6. Suppression of sclerostin by IL-1β may upregulate LRP6 phosphorylation and downstream of Wnt/β-catenin signaling is thought to be activated. In this condition, addition of sclerostin canceled the upregulation of Wnt/β-catenin signaling.

It is difficult to understand how the experiments were done, in particular, when the early effects were analyzed after treatment. For example, the results presented in Fig. 1C seem to be obtained after 3 weeks, but it is not clear when the cells were treated, and they represent early or late effect?

Response: We apologize inaccurate description regarding treatment of cells. This point was revised in the methods section (Line 74). The cells were treated with IL-1β and/or sclerostin for 3 weeks as the early stage experiment. In the present in vitro culture system, it takes at least 7 weeks to complete whole process of chondrogenic differentiation from mesenchymal chondroprogenitor to terminal calcification. The chondrogenic differentiation initiates to form cartilage nodules through a cellular condensation process that gave rise to proliferating chondrocytes and no hypertrophic chondrocytes are present by 3 weeks. After 5 weeks, mineralized regions began to appear in the preexisting cartilage nodules as tiny spots and they propagated throughout the cartilage nodules with time. Thus, the cells at 3 weeks were included in the early stage.

---

## [Editor Report · Decision Letter 2]

11 Sep 2020

Sclerostin inhibits interleukin-1β-induced late stage chondrogenic differentiation through downregulation of Wnt/β-catenin signaling pathway

PONE-D-19-33198R2

Dear Dr. Kumagai,

We’re pleased to inform you that your manuscript has been judged scientifically suitable for publication and will be formally accepted for publication once it meets all outstanding technical requirements.

Kind regards,

Michael Schubert

Academic Editor

PLOS ONE

---

## [Editor Report · Acceptance letter]

15 Sep 2020

PONE-D-19-33198R2

Sclerostin inhibits interleukin-1β-induced late stage chondrogenic differentiation through downregulation of Wnt/β-catenin signaling pathway

Dear Dr. Kumagai:

I'm pleased to inform you that your manuscript has been deemed suitable for publication in PLOS ONE. Congratulations! Your manuscript is now with our production department.

Kind regards,

on behalf of

Dr. Michael Schubert 

Academic Editor

PLOS ONE